

# Hard-bottom habitats support commercially important fish species: a systematic review for the North Atlantic Ocean and Baltic Sea

Hugo Flávio[1], Rochelle Seitz[2], David Eggleston[3], Jon C. Svendsen[4] and Josianne Støttrup[4]

[1] Wilfrid Laurier University, Waterloo, Ontario, Canada
[2] Virginia Institute of Marine Science, William & Mary, Gloucester Point, VA, USA
[3] Department of Marine, Earth & Atmospheric Sciences, North Carolina State University, Raleigh, NC, USA
[4] National Institute of Aquatic Resources, Technical University of Denmark, Lyngby, Denmark

## ABSTRACT

Hard-bottom habitats span a range of natural substrates (*e.g.*, boulders, cobble) and artificial habitats (*e.g.*, the base of wind turbines, oil platforms). These hard-bottom habitats can provide a variety of ecosystem services, ranging from the enhancement of fish biomass and production to providing erosion control. Management decisions regarding the construction or fate of hard-bottom habitats require information on the ecological functions of these habitats, particularly for species targeted in ecosystem-based fisheries management. This study provides a systematic review of the relationships of various hard-bottom habitats to individual commercially harvested species that are managed jointly across the Atlantic by the International Council for the Exploration of the Sea (ICES). We systemically reviewed peer-reviewed publications on hard-bottom habitats including both natural and artificial reefs, after applying various exclusion criteria. Most studies were conducted on near-shore hard-bottom habitats, and habitat importance varied according to fish species and region. We quantified the frequency with which studies demonstrate that natural and artificial hard-bottom habitats function as spawning grounds, settlement and nursery areas, and foraging grounds, as well as provide stepping-stones during migration, or new home ranges. Hard-bottom habitats generally support higher fish densities than surrounding habitat types, although not all fish species benefit from hard-bottom habitats. Of the commercially important species, cod (*Gadus morhua*) was the most frequently studied species, with enhanced biomass, density, feeding, and spawning on hard-bottom habitats compared to unstructured habitats. Moreover, hard-bottom habitats appear to be of particular importance for spawning of herring (*Clupea harengus*). Collectively, data indicate that loss of hard-bottom habitats may translate into less-favourable conditions for spawning and biomass of diverse commercial species, including cod and herring.

Corresponding author
Rochelle Seitz, seitz@vims.edu

## INTRODUCTION

Natural hard-bottom habitats are vital to the health and function of coastal ecosystems (*Stål, Pihl & Wennhage, 2007*; *Kovalenko, Thomaz & Warfe, 2012*; *Simon, Joyeux & Pinheiro, 2013*; *Lefcheck et al., 2019*). These structures provide a range of services, including structural refuge for juveniles and feeding grounds for commercially important fish species. Hard-bottom habitats include a wide variety of topographic forms (*e.g.*, high vertical profile, high complexity), substratum types (*e.g.*, boulders, rocks, cobble, artificial materials), and degree of rugosity (*e.g.*, smooth surfaces, perforated materials), with well-documented effects of increasing biodiversity with habitat complexity (*Wilhelmsson, Yahya & Öhman, 2006*; *Kovalenko, Thomaz & Warfe, 2012*; *Wehkamp & Fischer, 2013*). As such, these hard substrates often comprise a three-dimensional structure, which means they can offer increased complexity in an otherwise two-dimensional landscape. Hard-bottom habitats provide substrate for the attachment of vegetation, such as kelp and other macroalgae that further increase the complexity of the benthos and associated biodiversity, compared to unstructured habitat (*Norderhaug et al., 2005*; *Christie, Jørgensen & Norderhaug, 2007*; *Christie, Norderhaug & Fredriksen, 2009*). Reviews on the importance of coastal habitats for exploited species identified a lack of information on how fish utilize some habitat types in the North Atlantic, particularly complex natural and artificial hard-bottom habitats (*e.g.*, rocky shores, foundations of wind farms, oil-rigs; *Seitz, 2014*; *Seitz et al., 2014*; *Fowler et al., 2018*).

Artificial reefs have been used as a tool for restoration of marine habitats and as mitigation for damage to other marine habitats. Artificial reefs may be constructed using natural stone (*Støttrup et al., 2017*) or, more commonly, using a range of man-made materials including concrete units or scrap materials, such as decommissioned ships and oilrigs (*Jensen, 2002*; *Soldal et al., 2002*). Artificial reefs have been widely deployed to (1) enhance fisheries (*Seaman & Sprague, 1991*; *Bombace et al., 1994*; *Brock, 1994*), (2) mitigate damage to specific bottom habitats such as seagrass beds (*Posidonia* sp.; *González-Correa et al., 2005*), (3) increase the recreational value of an area (*Wilhelmsson et al., 1998*) or (4) as a tool for rehabilitation of coastal ecosystems (*Pickering, Whitmarsh & Jensen, 1999*). Artificial reefs can be intentional (*i.e.,* anthropogenic explicit creation of a reef) or unintentional (*i.e.,* created as a by-product of other anthropogenic activities, such as oil platforms and wind turbines, hereafter referred to ''*de facto*'' reefs). Artificial reefs generally harbor higher fish densities and biomass than surrounding natural reefs (*Wilhelmsson et al., 1998*), which is attributed to their structural complexity (*Hunter & Sayer, 2009*; *Degraer et al., 2020*) and/or food abundance (*Rilov & Benayahu, 2000*; *Fabi, Manoukian & Spagnolo, 2006*; *Glarou, Zrust & Svendsen, 2020*), which may also increase abundance and richness of the fish community (*Rilov & Benayahu, 2000*). Moreover, effects of artificial reefs may vary by individual fish species (*Leitão et al., 2008*; *Krone et al., 2013*). Findings on the importance of artificial habitats for species managed by the International Council for the Exploration of the Sea (ICES) could be important for management decisions regarding habitat restoration and protection of fisheries species.

In Europe, hard-bottom habitats are one of the few marine habitat types included in the European Union's Habitat Directive (1170 Reef Habitat; *EEC, 1992*). For this reason, reef areas are included in several Natura 2000 networks such as the Danish Natura 2000 network. Surprisingly, the importance of temperate reefs is rarely considered in fisheries management and in fish population assessments (*Lipcius et al., 2019*; and references therein), and the importance of hard-bottom reefs may differ for different fishery species. Management decisions require the best available information on specific ecosystem services that, in some cases, can be monetized (*Costanza et al., 1997*; *Cavanagh et al., 2016*). As such, the goal of this study was to summarize the existing information on ICES-managed fish species and their use of temperate hard-bottom habitats in an effort to (1) inform scientists interested in the ecological function of these habitats for some key fish species, and (2) inform managers (especially ICES managers) and conservation groups focused on coastal habitats of the potential importance of hard-bottom habitats to certain fish species. ICES is an inter-governmental marine science organization whose advisory committee translates science to management on "the sustainable use and protection of marine species," and their work is relevant to extensive coastal areas, for example throughout Europe and the United States. Focusing our review on ICES-managed species is important because ICES-managed species are common throughout coastal areas of North America and Europe, and many commercial and recreational fisheries for these species are in poor condition, which causes concerns among conservationists as well as those dependent on fisheries as an economic activity or food source (*International Council for the Exploration of the Sea , 2013*).

## MATERIALS & METHODS

The overarching objective was to review the literature to quantify the ways in which hard-bottom habitats support commercially important fish species in temperate regions, particularly the North Atlantic, Baltic, and North Seas. We targeted the value of natural hard-bottom habitats and artificial reefs on exploited fish species by (1) summarizing increases or decreases in demographic variables (*e.g.*, growth, survival, biomass, spawning rates) for individual commercial fishery species associated with hard-bottom habitats, and (2) highlighting the importance of these habitats for certain fish species so that managers and conservation groups can act on this knowledge. The spatial focus was on temperate, hard-bottom habitats (excluding subtropical and tropical areas, and the Mediterranean Sea), and we excluded some natural hard-bottom habitats such as coral reefs and biogenic habitats (see negative keywords in the search string below). We also focused on species important for management by ICES (see species-related keywords below), therefore, we did not include all commercially important species. In addition, we excluded rockfish in the Pacific Ocean, which have a separate body of literature targeted towards them, and which are not managed by ICES. Thus, our conclusions cannot be applied to all species in every area, and our focus was on non-biogenic or anthropogenically created reefs. We were most interested in habitat enhancement from the physical structure provided by hard-bottom habitats, and thus excluded biogenic reefs, which provide additional ecological benefits such as increased nutrients (*e.g.*, from faeces and pseudofaeces on mussel beds; *Thiel & Ullrich, 2002*).

## Search protocol

We followed the protocol for a systematic literature review proposed by *Pullin & Stewart (2006)*, which set forth guidelines for planning the review and analysis of the data. We systematically reviewed the literature and evaluated articles returned by three different databases (DTU-Findit, Web of Science, and Scopus) for relevance based on our inclusion criteria (Table 1). Screening was conducted at the level of title, abstract, and full text (*Pullin & Stewart, 2006*) to each of several ecological study variables (see "search string" below), using the PRISMA protocol (*Moher et al., 2009*) (Fig. 1). The first author of this publication screened each article by title to eliminate non-relevant articles, and a subset of articles was sent to all co-authors to compare the group's decisions and ensure we came to consensus on what should be discarded. Then, the same procedure was followed for examining the abstracts of the remaining articles. Finally, we divided the articles for full-text examination among all authors and each person independently extracted data, with a group discussion of which articles were relevant and why, and resolving any discrepancies on what should be included. The search was run on the three different databases on April 7th, 2017. The articles returned by the three different databases were evaluated for relevance based on the inclusion criteria (Table 1), applied at three successive levels: (1) title, (2) abstract and (3) full-text. It is important to note that there are some limitations of our approach, and that we may have missed some research that did not fall into our selected categories based on the search terms we used, and results are only applicable to the species we incorporated into our search (namely ICES-managed species). Our review focused on the species for which ICES gives advice, which resulted in examining the species of importance for ICES Member Countries (*i.e.,* Belgium, Canada, Denmark, Estonia, Finland, France, Germany, Iceland, Ireland, Latvia, Lithuania, the Netherlands, Norway, Poland, Portugal, Russia, Spain, Sweden, United Kingdom, and United States of America; US and Canadian fish stocks are not included in the advice, though these are ICES Member Countries). Therefore, interpretations of these data apply only to the species and habitats we incorporate. Additionally, we examined the current literature (2017–2022) for more recent articles on the same ICES-managed species using hard bottom habitats (database from B Ciotti pers. comm., 2022), which allowed us to identify a couple of case studies to serve as a check on validity of conclusions from our systematic search.

### Search string

The following search keywords were chosen for each of the ecological variables we wanted to examine: Biomass, Density, Biodiversity, Condition, Mortality, Growth, Feeding, Spawning, Nursery, Settling, Diel cycle, Site fidelity, Migration. Separate searches with all other keywords in combination with each of the ecological search keywords were conducted (Table 2). Location-related keywords and exclusion keywords were also included to narrow the search results.

**Importance-related keywords** (note: asterisks represent a search engine wild-card and were used to broaden the search by finding multiple words that start with the same letters):

**Table 1  Admission/exclusion criteria.** Admission/exclusion criteria for hard-bottom review.

| Criteria | Include | Exclude |
|---|---|---|
| Peer-reviewing | Peer-reviewed | Everything else |
| Text Language | English | Everything else |
| Ecosystem | Natural and artificial hard-bottom reefs in marine or contiguous salt water environments (*i.e.*, fjords and estuaries) | Biogenic reefs or other structures placed at non-marine environments |
| Target species | Fish species managed by ICES | No fish species managed by ICES present |
| Location | North hemisphere, temperate zone (not including Mediterranean Sea) | Mediterranean Sea, everything else |
| Type of study | Field studies | Everything else |

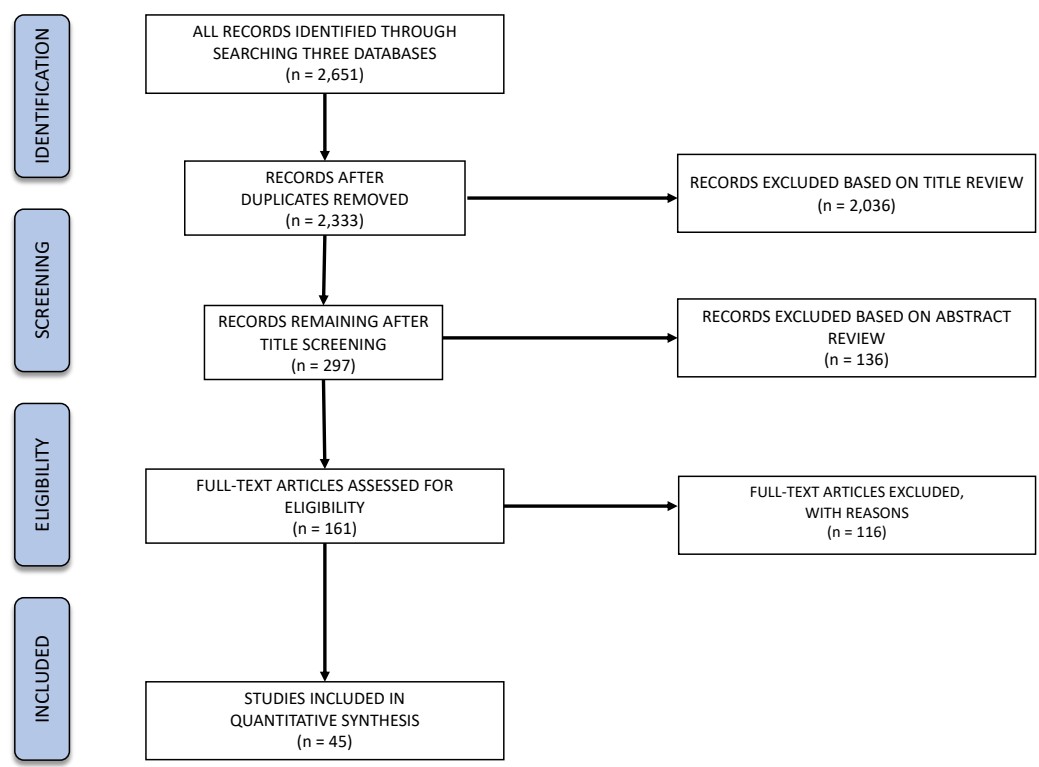

**Figure 1  PRISMA flow diagram for our hard-bottom systematic review showing numbers of articles at different stages in the review.**

Fish attraction, fish production, fish biomass, fish diversity, fish growth, fish survival, fish density, fish migration, demographic rate*, fish reproduction, fish mortality and fish juvenile*, feeding, nursery, spawn*.

**Ecosystem-related keywords:** Hard bottom, rocky reef*, stone reef*, riprap, seawall*, groin*, groyne*, offshore wind energy structure*, wind farm*, wind turbine*, artificial reef*, fish attraction device*, FAD*, shipwreck*, oil rig*, oil platform* oil pipe*, gas rig*, gas platform*, gas pipe*, offshore platform*, rig* to reef*, decommission*.

**Table 2** **Total number of articles (no parentheses) and percentage of articles in relation to the total of articles (in parentheses) for each of the top eight fish species that were most commonly selected in our review, in relation to the different ecological variables of interest in the review.** The groups of variables are organized into the following order: community-, fitness-, reproduction- and distribution-related (from top to bottom; terms defined in the Materials & Methods section). In the "Total" column, the percentage of articles is shown in relation to the total number of articles reviewed (*i.e.*, 45). Note that any given article may simultaneously target more than one species and/or variable. Most research has targeted few species (*i.e.*, cod, herring, and saithe) and a limited set of variables (*i.e.*, biomass, density, biodiversity, and feeding). Full genus names given in "Species-related keywords" section.

| | *G. morhua* | *C. harengus* | *P. virens* | *M. surmu.*[a] | *M. merlangus* | *P. pollachius* | *P. platessa* | *S. solea* | **Total** |
|---|---|---|---|---|---|---|---|---|---|
| Biomass | 3(12) | 1(8.3) | 1(9.1) | 3(30) | 3(30) | 2(22.2) | 1(11.1) | 0(0) | 6(13.3) |
| Density | 15(60) | 6(50) | 5(45.5) | 7(70) | 7(70) | 6(66.7) | 7(77.8) | 5(62.5) | 24(53.3) |
| Biodiversity | 6(24) | 4(33.3) | 5(45.5) | 7(70) | 6(60) | 5(55.6) | 5(55.6 | 7(87.5) | 15(33.3) |
| Condition | 3(12) | 0(0) | 0(0) | 0(0) | 0(0) | 0(0) | 0(0) | 0(0) | 3(6.7) |
| Mortality | 2(8) | 2(16.7) | 0(0) | 0(0) | 0(0) | 0(0) | 0(0) | 0(0) | 3(6.7) |
| Growth | 1(4) | 0(0) | 1(9.1) | 0(0) | 0(0) | 0(0) | 0(0) | 0(0) | 2(4.4) |
| Feeding | 8(32) | 2(16.7) | 4(36.4) | 1(10) | 1(10) | 2(22.2) | 2(22.2) | 0(0) | 12(26.7) |
| Spawning | 1(4) | 4(33.3) | 0(0) | 0(0) | 0(0) | 0(0) | 0(0) | 0(0) | 5(11.1) |
| Nursery | 0(0) | 0(0) | 0(0) | 0(0) | 0(0) | 0(0) | 0(0) | 0(0) | 0(0) |
| Settling | 1(4) | 0(0) | 0(0) | 0(0) | 0(0) | 0(0) | 0(0) | 0(0) | 1(2.2) |
| Diel cycle | 4(16) | 1(8.3) | 2(18.2) | 1(10) | 1(10) | 2(22.2) | 1(11.1) | 1(12.5) | 4(8.9) |
| Site fidelity | 3(12) | 0(0) | 0(0) | 0(0) | 0(0) | 0(0) | 0(0) | 0(0) | 3(6.7) |
| Migration | 1(4) | 0(0) | 0(0) | 0(0) | 0(0) | 0(0) | 0(0) | 0(0) | 1(2.2) |
| Total | 25(55.6) | 12(26.7) | 11(24.4) | 10(22.2) | 10(22.2) | 9(20) | 9(20) | 8(17.8) | 45 |

**Notes.**
[a]*Mullus surmuletus.*

**Species-related keywords:** *Alopias* spp., *Amblyraja radiata*, *Ammodytes* spp., *Anguilla anguilla*, *Aphanopus carbo*, *Argentina silus*, *Beryx* spp., *Brosme brosme*, *Capros aper*, *Centrophorus squamosus*, *Centroscymnus coelolepis*, *Cetorhinus maximus*, *Chelidonichthys cuculus*, *Clupea harengus*, *Coryphaenoides rupestris*, *Dalatias licha*, *Dicentrarchus labrax*, *Dipturus batis-complex*, *Engraulis encrasicolus*, *Gadus morhua*, *Galeorhinus galeus*, *Galeus melastomus*, *Hoplostethus atlanticus*, *Lamna nasus*, *Lepidorhombus* spp., *Leucoraja circularis*, *Leucoraja fullonica*, *Leucoraja naevus*, *Limanda limanda*, *Lophius budegassa*, *Lophius piscatorius*, *Macrourus berglax*, *Mallotus villosus*, *Melanogrammus aeglefinus*, *Merlangius merlangus*, *Merluccius merluccius*, *Micromesistius poutassou*, *Molva dypterygia*, *Molva molva*, *Mullus surmuletus*, *Mustelus* spp., *Pagellus bogaraveo*, *Phycis blennoides*, *Platichthys flesus*, *Pleuronectes platessa*, *Pollachius pollachius*, *Pollachius virens*, *Raja brachyura*, *Raja clavata*, *Raja microocellata*, *Raja montagui*, *Raja undulata*, *Reinhardtius hippoglossoides*, *Rostroraja alba*, *Salmo salar*, *Salmo trutta*, *Sardina pilchardus*, *Scomber scombrus*, *Scophthalmus maximus*, *Scophthalmus rhombus*, *Scyliorhinus canicula*, *Scyliorhinus stellaris*, *Sebastes* spp., *Solea solea*, *Sprattus sprattus*, *Squalus acanthias*, *Squatina squatina*, *Trachurus picturatus*, *Trachurus trachurus*, *Trachyrincus scabrus*, and *Trisopterus esmarkii*.

**Location-related keywords:** Belgium, Canada, Denmark, Estonia, Finland, France, Germany, Iceland, Ireland, Latvia, Lithuania, Netherlands, Norway, Poland, Portugal, Russia, Spain, Sweden, United Kingdom, United States of America, Northeast Atlantic, Atlantic Iberian, Biscay, English Channel, Bristol Channel, North Sea*, Irish Sea*, Cantabrian Sea*, Belt Sea*, Bothnian Sea*, Celtic Sea*, Azores, Skagerrak, Kattegat,

Bornholm, Baltic, Gulf of Gdansk, Gulf of Riga, east Gotland, Faroe grounds, Iceland grounds, Greenland, Rockall, Bothnian bay, Reykjanes Ridge, Dogger bank⋆, Viking bank⋆, Bergen bank⋆.

**Negative (Excluded) keywords:** Biogenic reef⋆, eelgrass, mussel bed⋆, coral reef⋆, tropical, subtropical, Australia, Mediterranean, Caribbean, Indian Sea.

### Data extraction

Our primary question was: "What is the habitat importance of hard-bottom natural and artificial structures for temperate fish species managed by ICES?" To answer this question, for each article we determined the studied habitat, the ecosystem type, and species targeted, as detailed below. We broke down habitat importance into the following four main components: (1) distribution-related variables, (2) community-related variables, (3) fitness-related variables, and (4) reproduction-related variables (see details below).

#### Habitat importance

Fish species receive numerous benefits from benthic habitats, such as food, shelter, or spawning grounds. Such benefits may translate into measurable demographic variables (*e.g.*, growth, survival, biomass, spawning rates), which allow researchers to assess the importance of various habitats. We initially grouped the data into the following four broad categories: (1) species distribution, (2) community, (3) fitness, and (4) reproduction. Distribution-related variables included articles targeting diel cycles, site fidelity, or migration patterns. Community-related variables included articles targeting fish biomass, density, or biodiversity. Fitness-related variables included articles targeting condition, mortality rates, growth, and feeding. Lastly, reproduction-related variables included articles targeting spawning habitats, nursery potential, and settling (both of adults to newly formed habitats or of juveniles).

#### Ecosystem type

A species' ecological niche may encompass different ecosystems. Therefore, we targeted coastal marine environments and transitional areas, such as fjords and estuaries. Though we excluded biogenic reefs, we did include literature that dealt with macrophyte cover in temperate regions.

#### Target species

To improve future fisheries management, this review included fish species targeted for advice by the International Council for the Exploration of the Sea (ICES). Targeted species are listed above in "Species-related keywords."

## RESULTS & DISCUSSION

Our study indicates that hard-bottom substrates may benefit multiple fish groups, such as elasmobranchs (*e.g.*, *Martin et al., 2012*; *Rodríguez-Cabello et al., 2008*), gadoids (*e.g.*, *Reubens, Degraer & Vincx, 2014*; *Ross, Rhode & Quattrini, 2015*; *van Hal, Griffioen & van Keeken, 2017*), and other teleosts such as seabass (*Dicentrarchus labrax*; *Leitão et al., 2008*). Furthermore, hard substrates serve as: (1) stepping stones, where species may find shelter

during migration events (*Gascon & Miller, 1982*), (2) spawning grounds for demersal eggs (*Johannessen, 1986*), (3) settlement grounds for young fish migrating out of nurseries (*Santos, Monteiro & Lasserre, 2005*),  (4) foraging areas with increased prey densities (*Leitão et al., 2008*) or (5) new home ranges, with newcomers developing high site fidelity rates, as observed for novel constructions such as marine wind farms (*Reubens et al., 2013*).

## Review selection process

The search process returned an initial 2,651 articles, paring down to 2,333 articles after exclusion of duplicate references. Of those, 45 articles were returned after the subsequent screening criteria and were examined for full-text data extraction (as detailed below and in Fig. 1).

### *Title analysis*

The title reviewing process led to the exclusion of 2,036 articles (87% of the initial 2,333), leaving 297 articles for abstract analysis. Most of the exclusions (54%) represented off-topic articles dealing with, for example, energy production, aquaculture, human health, oil spills, or physiology. Additionally, 35% of the excluded articles were explicitly related to non-target organisms (*e.g.*, zebrafish, rainbow trout, invertebrates). A total of 9.5% excluded articles that either studied non-target locations or other ecosystems (*e.g.*, Hawaii or soft-bottom ecosystems, respectively). Lastly, 1.4% of the excluded titles either did not correspond to peer-reviewed articles or were not written in English.

### *Abstract analysis*

From the 297 analysed abstracts, a total of 136 failed to comply with the inclusion criteria (46%), leaving 161 articles for full text analysis. Most of the exclusions represented articles related to non-target species (32%), followed by articles that did not relate to the target topics (25%). A total of 30% excluded articles either studied non-target locations or other ecosystems. Additionally, 13% of the analysed search results were not from the peer-reviewed literature.

### *Full text analysis*

From the 161 full texts analysed, 116 (72%) did not comply with the exclusion criteria. Most of the articles were excluded based on location (49%), corresponding mostly to studies targeting rockfish in the Pacific Ocean. A total of 21% exclusions represented articles related to non-target species, while 13% studied non-target ecosystems and 7% did not relate to the target topics. A total of 11 articles were excluded for "other" reasons: four were not peer-reviewed, four were reviews, and for three it was not possible to obtain a full text version of the document. Therefore, the review focused on 45 studies distributed along the Baltic and North Seas and northern Atlantic Ocean (Fig. 2). We quantified the numbers of articles or percentage of articles reviewed as a function of the various response variables emphasized.

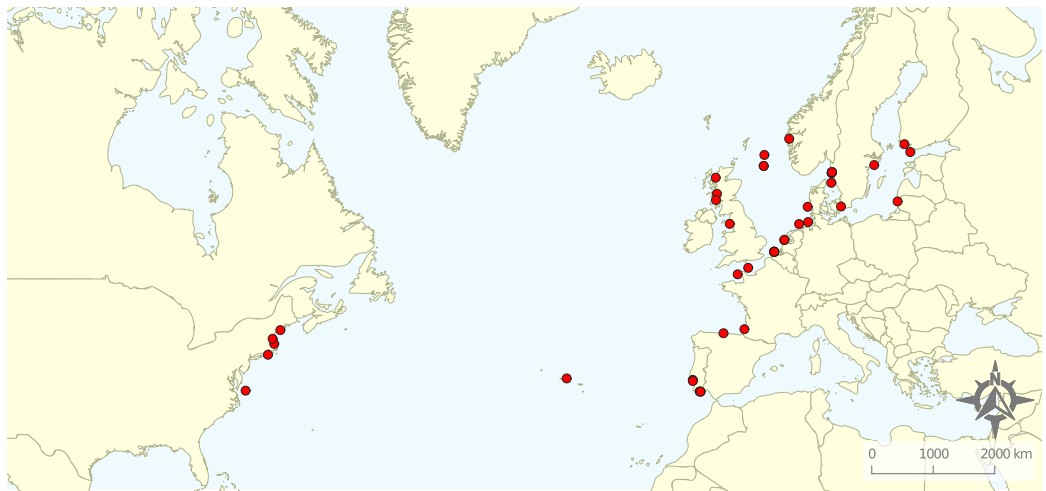

**Figure 2** **Geographical distribution of the articles that were included in the review.**

## Articles reviewed as a function of response variables emphasized

Distribution-related variables (*i.e.,* diel cycle, site fidelity, migration) were studied in six articles (13.4%). These targeted primarily site fidelity and migration patterns of Atlantic cod (*Gadus morhua*; *e.g.,* *Reubens et al., 2014*; *Reubens, Degraer & Vincx, 2014*).

Community-related variables (*i.e.,* density, biodiversity, biomass) were the most frequently studied amongst the articles reviewed, being analysed in two thirds of the studies (Fig. 3). Most research targeted a few species (*i.e.,* Atlantic cod, herring (*Clupea harengus*), and saithe (*Pollachius virens*)) and a limited set of variables (*i.e.,* biomass, density, biodiversity, and feeding). Cod was the species most commonly studied in community-related (*i.e.,* density, biodiversity, biomass) studies (Table 2).

Fitness-related variables (*i.e.,* condition, mortality, growth, feeding) were studied in 17 articles (38% of the total number of analysed articles). Feeding was studied primarily on cod and saithe, showing a tendency for fish use of hard-bottom habitats for foraging (Fig. 4; *e.g.,* *Johannessen, 1986*; *Malek, Collie & Taylor, 2016*; *Wennhage & Pihl, 2002*). Interestingly, with the exception of cod, little research has been carried out on the remaining fitness-related variables (Table 2).

Reproduction-related variables (*i.e.,* spawning, settling, nursery) were studied in six articles (13.4%). Spawning was studied primarily on herring and cod, with articles on nursery and settling being almost absent for other species in the reviewed literature (Table 2). From the species targeted in our review, hard-bottom substrates appeared to be of particular importance for herring spawning (*Aneer & Nellbring, 1982*; *Johannessen, 1986*; *Käariä et al., 1997*; *Šaškov et al., 2014*).

There was a clear predominance in the study of community-related variables in the analysed research (Fig. 3). Broadening the scope of studied variables and targeted species may prove essential to fully understand the relationships between hard-bottom reefs and commercially important species. For example, more quantitative research at the community level reveals the value of hard-bottom habitats for fish populations (essential

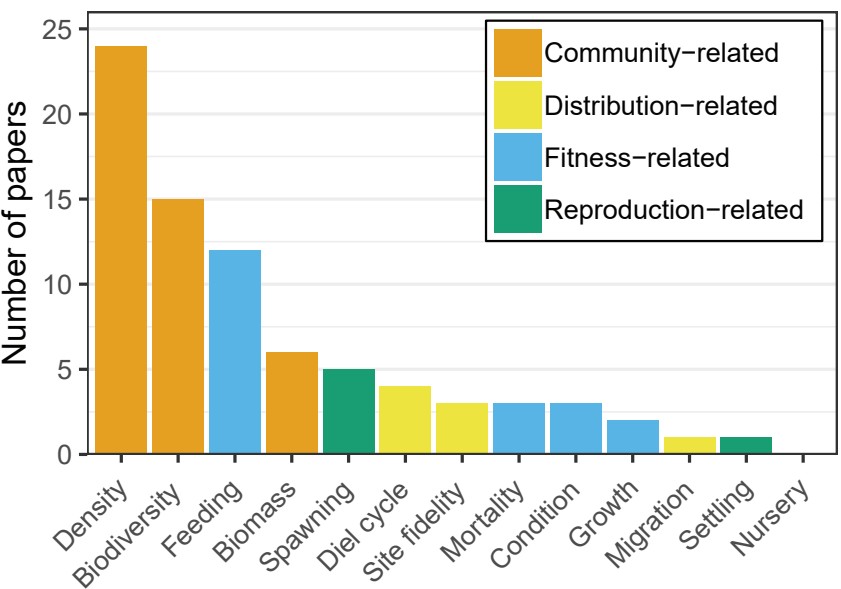

**Figure 3** **Number of articles targeting each of the ecological response variables recorded.** Note that each article may target more than one variable. Variables were grouped by colors, showing a clear dominance of community-related studies.

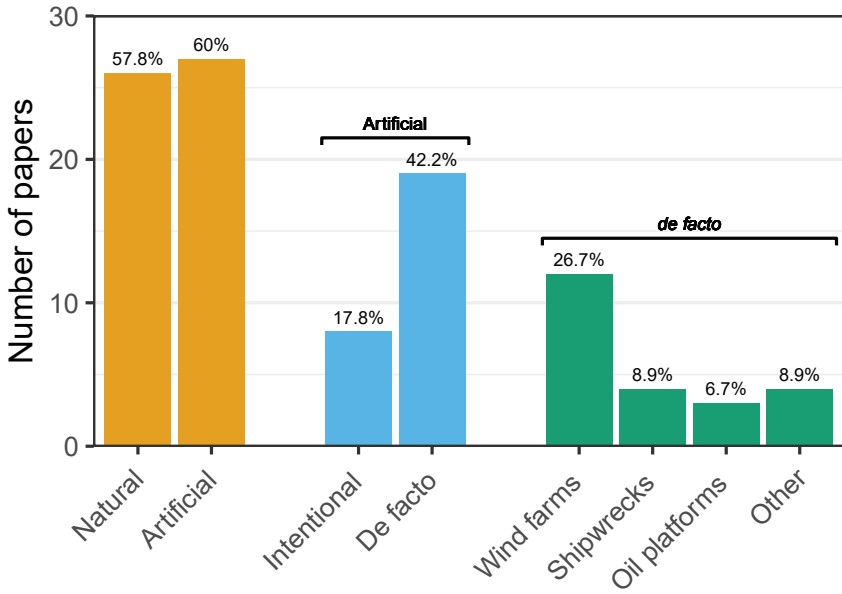

**Figure 4** **Number of articles targeting each of the studied hard-bottom substrates.** Note that articles may target more than one topic simultaneously.

fish habitat, *sensu Valavanis, 2008*; or effective habitat, *sensu Dahlgren et al., 2006*), and integrating this information into assessment and management may provide considerable advances towards ecosystem-based fisheries management (*e.g.,* fish production estimates that can be monetized).

## Ecosystems explored

Research was nearly equally distributed across natural and artificial reef structures, with 26 and 27 articles related to each, respectively, and eight targeting both natural and artificial structures simultaneously (Fig. 4, Table 3). Furthermore, 21 articles (46.7%) that resulted from our search performed comparisons of hard-bottom habitats with soft-bottom habitats. The review indicated that different regions of the world have different research foci. All articles on the north-western shore of the Atlantic Ocean focused on natural reefs, with one article also targeting an intentional artificial reef (Fig. 5). Similarly, there was a greater focus on natural reef research in the Baltic Sea, with nine out of eleven articles targeting natural reefs, as well as three articles targeting artificial *de facto* reefs, and only one article targeting intentional artificial reefs. Conversely, in the North Sea, only three out of 15 articles targeted natural reefs, while 14 targeted *de facto* artificial reefs (most of which explored the effects of wind farms). Finally, on the north-eastern coast of the Atlantic Ocean, there were nearly equal numbers of articles on natural and artificial reef research (six and five articles, respectively, plus two articles that targeted both reef types simultaneously). However, of the articles targeting artificial reefs, only one studied *de facto* artificial reefs, while the remaining explored artificial intentional artificial reefs. There was a noticeable paucity of studies in the northern regions. Additionally, most of the analysed research has been conducted near-shore. Broadening the scientific knowledge on the importance of different habitats, areas, and regions along a depth gradient away from shore may provide crucial insights for future management.

## Frequently studied species

There has been a consistent research effort towards multiple morphological and behavioural aspects of cod, with this species being researched twice as frequently as herring, the second-most-researched species (25 *vs* 12 articles, respectively; Table 2). Furthermore, while research on cod and herring is spread across the four groups of ecological variables considered in our review (*i.e.,* species distribution, community, fitness, reproduction), research is lacking for most reviewed species on variables not related with community (*e.g.,* condition, settling, site fidelity; Table 2).

The presence of hard-bottom habitat was associated with enhanced biomass in cod and saithe and, conversely, was negatively associated with biomass in herring and whiting (*Merlangius merlangus*; Fig. 6). Most articles compared fish densities in multiple habitats, and higher densities of cod on hard-bottom habitats were demonstrated in several articles, with only one article suggesting fewer cod on hard-bottom compared to adjacent soft-bottom habitats. Both positive and negative relationships of hard-bottom habitats with density were demonstrated for herring, saithe, striped red mullet (*Mullus surmuletus*), whiting, pollock (*Pollachius pollachius*), plaice (*Pleuronectes platessa*), and sole (*Solea*
**Table 3 Distribution of articles per fish species and different hard-bottom structures.** Artificial reefs are further split into intentional and *de facto* reefs, with *de facto* reefs further subdivided into the various structures that form *de facto* reefs. "Other" includes wave-energy foundations and coastal-protection structures (*i.e.*, riprap and breakwater tetrapods). The percentage of articles in relation to the total of articles for each species is shown in the brackets. In the "Total" column and row, the proportion of articles is shown in relation to the total number of articles reviewed (*i.e.*, 45). Note that any given article may simultaneously target more than one species and/or more than one hard-bottom structure. Overall, data show an even distribution of research between natural and artificial reefs, despite the spatial patterns noted in Fig. 4. Full genus names given in "Species-related keywords" section.

| | G. morhua | C. harengus | P. virens | M. surmu.[a] | M. merlangus | P. pollachius | P. platessa | S. solea | Total |
|---|---|---|---|---|---|---|---|---|---|
| Natural Reefs | 12(48) | 9(75) | 9(81.8) | 5(50) | 5(50) | 7(77.8) | 5(55.6) | 3(37.5) | 26(57.8) |
| Artificial Reefs | 18(72) | 4(33.3) | 7(63.6) | 6(60) | 7(70) | 4(44.4) | 6(66.7) | 7(87.5) | 27(60) |
| Intentional | 3(12) | 1(8.3) | 3(27.3) | 2(20) | 1(10) | 3(33.3) | 1(11.1) | 2(25) | 8(17.8) |
| *De facto* | 15(60) | 3(25) | 4(36.4) | 4(40) | 6(60) | 1(11.1) | 5(55.6) | 5(62.5) | 19(42.2) |
| Wind Farms | 10(40) | 3(25) | 1(9.1) | 3(30) | 6(60) | 0(0) | 5(55.6) | 5(62.5) | 12(26.7) |
| Shipwrecks | 3(12) | 0(0) | 1(9.1) | 1(10) | 1(10) | 0(0) | 1(11.1) | 0(0) | 4(8.9) |
| Platforms | 2(8) | 0(0) | 3(27.3) | 0(0) | 0(0) | 0(0) | 0(0) | 0(0) | 3(6.7) |
| Other | 3(12) | 0(0) | 0(0) | 1(10) | 0(0) | 1(11.1) | 0(0) | 0(0) | 4(8.9) |
| Total | 25(55.6) | 12(26.7) | 11(24.4) | 10(22.2) | 10(22.2) | 9(20) | 9(20) | 8(17.8) | 45 |

**Notes.**
[a] *Mullus surmuletus.*

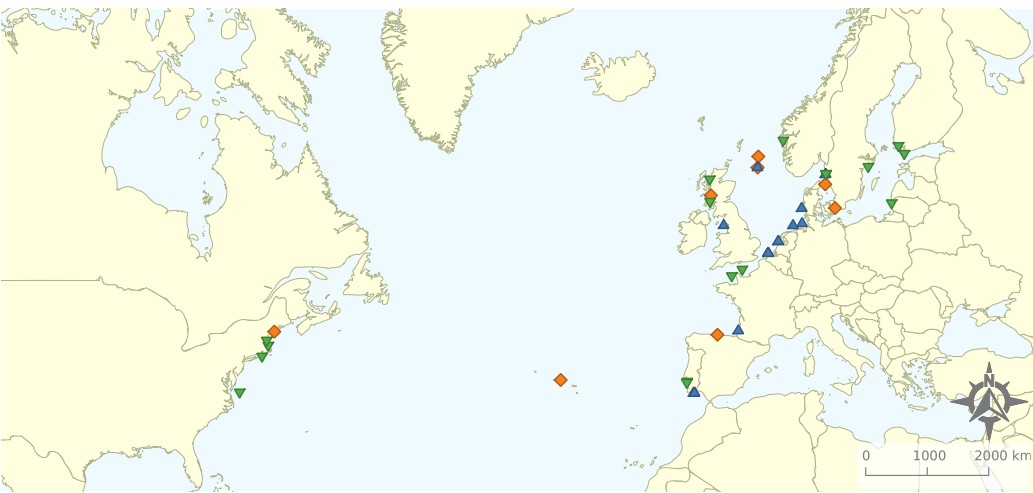

**Figure 5 Distribution of articles with grouping of studied reef structures.** Natural reefs only (downward pointing green triangles); artificial reefs only (upward pointing blue triangles); both natural and artificial reefs (orange diamonds).

*solea*). Cod, herring, saithe, striped red mullet, whiting, and pollock appeared to benefit from foraging on hard-bottom habitat (*i.e.,* for plaice feeding, only negative relationships of hard-bottom habitats were reported, as plaice feed less frequently over hard-bottom habitats). Few articles targeted spawning on hard-bottom habitats, and those that did reported associated benefits for cod and herring.

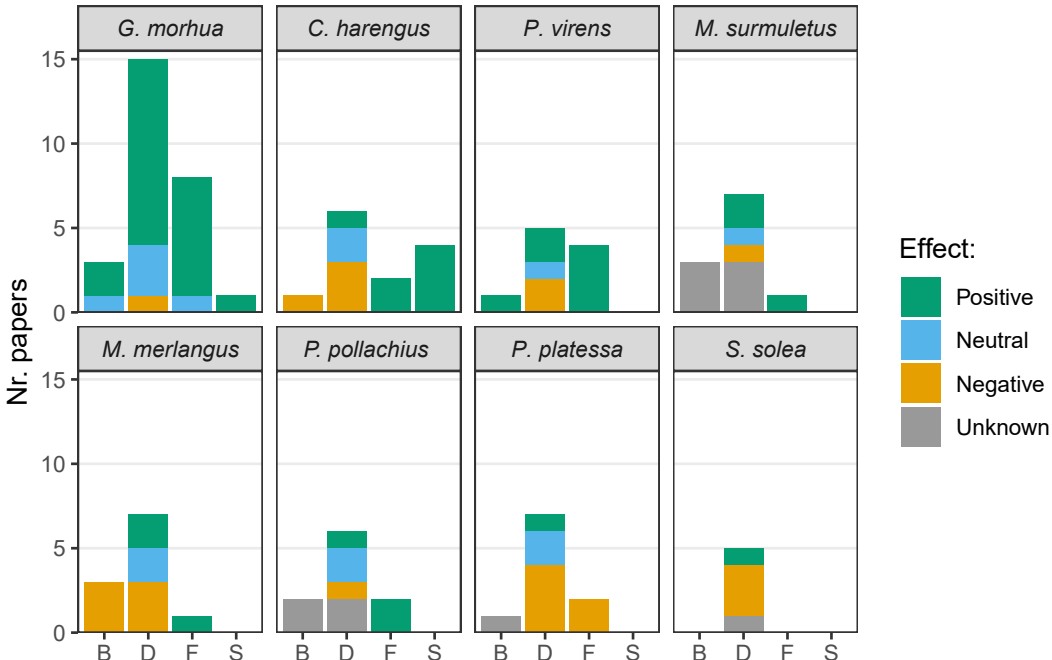

**Figure 6** **Effects of the presence of hard-bottom habitats on the most commonly studied variables.** Biomass (B), density (D), feeding (F) and spawning (S), for the species most commonly targeted in recent research. Given the variety of methods used in literature the effects are listed as positive, neutral or negative. When no comparison was performed with a different substrate, the effect was considered unknown.

## Changes in fish distribution

Research has demonstrated positive effects of the presence of hard-bottom substrates on cod biomass and density (*e.g.*, *Krone et al., 2013*; *Pihl & Wennhage, 2002*). With a few exceptions, natural hard-bottom habitats and artificial reefs harbor significantly higher abundances of fish than adjacent unstructured habitats (*Rodríguez-Cabello et al., 2008*; *Fujii, 2016*). Fish movement patterns reflected relatively high site fidelity to structures (*e.g.*, *Hartman, 1987*; *Reubens et al., 2014*; *Reubens, Degraer & Vincx, 2014*), and the stomach contents of fish residing on reefs (excluding biogenic reefs) reflected the prey assemblages associated with the reefs (*e.g.*, (*Reubens et al., 2014*)), suggesting extended residency and foraging in the reef areas. Interestingly, catch per unit effort (CPUE) of cod was greater adjacent to wind farms than in shipwrecks in the Belgian part of the North Sea (*Reubens, Degraer & Vincx, 2014*), suggesting attraction towards wind farms. It is important to note that the studied wind farms were closed to fisheries, while the shipwreck sites had no fisheries restrictions, possibly biasing the comparison.

Gadoids such as saithe, haddock (*Melanogrammus aeglefinus*) and cod were the most common species at the "Miller" oil and gas platform in the North Sea (*Fujii, 2016*). Cod abundance was approximately four times higher at the platform than in nearby shipwrecks or sandy bottom. Using trawl data, *Martin et al. (2012)* modeled distributions of ten demersal elasmobranchs in the eastern English Channel in relation to the environment,

with female small-spotted catshark (*Scyliorhinus canicula*) occurring at higher densities on hard than soft substrate.

Within four years after the deployment of artificial reefs in an area previously closed to trawl fisheries in the Bay of Biscay, there was a rapid increase in the biomass of small-spotted catshark (*Rodríguez-Cabello et al., 2008*). Importantly, four to eight years after deployment, axillary sea-bream (*Pagellus acarne*), horse mackerel (*Trachurus trachurus*), and striped red mullet demonstrated significant increases in biomass. The observation that horse mackerel responded positively to the addition of artificial reefs demonstrates that schooling pelagic species may also benefit from these hard-bottom habitats.

Reefs (excluding biogenic reefs) may also enhance economic returns. For example, in one location, reefs near Algarve and Faro, Portugal, raised catch rates and economic returns (*Neves Santos & Costa Monteiro, 1998*). Specifically, CPUE at the reefs exceeded CPUE at the control sites by a factor of 2.03–2.28 for the protected reef and by a factor of 1.11–1.86 for the exploited reef. In some artificial reef systems, the economic return per unit of effort was higher at the reefs than at the control sites. Fishing on artificial reefs would be expected to yield 2.2 times the economic return compared to the control sites (*Neves Santos & Costa Monteiro, 1998*). The productivity of reefs increased throughout the 15-year period, implying that the interaction between the hard substrate and the marine environment may take many years to produce the full effects on the carrying capacity (*Whitmarsh et al., 2008*). A distinct spatial pattern of fishing activity around the reefs was detected, with a concentration of fishing gear on the periphery of the reef area and a lower gear density at farther distances. A concentration of fishing gear nearby the reef would suggest that the improved fishing opportunities offered by the reefs have been recognized by fishermen. Fishermen likely take advantage of the spill-over of fish from the reefs, consistent with recent studies (*Rosemond et al., 2018*).

The inclusion of quantitative data on hard-bottom habitats in the monitoring of fish populations may further resolve a mismatch between fish population assessments and catches in fisheries of several fish species with a patchy distribution due to their high affinity for hard-bottom habitats (*e.g.*, cod; *Wieland et al., 2009*). The mismatch between fish population assessments and catches of fish species with a patchy distribution may be further exacerbated during stock recovery, where the most suitable habitats (*i.e.,* hard bottoms) will be populated first according to the ideal free distribution theory of *MacCall (1990)*.

Not all fish species benefit from hard-bottom or marine constructed structures. Neutral or negative relationships with biomass and density were recorded for flatfish such as plaice (*Pleuronectes platessa*) and sole (*Solea solea*), as well as for whiting (*Merlangius merlangus*). For example, *Støttrup et al. (2014)* found decreasing densities of sole post restoration of a natural reef, and *Bergström, Sundqvist & Bergström (2013)* only found whiting before the implementation of a wind farm. Neutral or negative relationships with hard-bottom habitat for some species reflects the preferred habitat of the different species, with both flatfish and whiting occurring primarily on soft-bottom habitats (*Stål, Pihl & Wennhage, 2007*).
The rapid growth of offshore wind farms, coupled with the high abundance and biomass of fish associated with wind farms, suggest that these *de facto* marine protected areas (MPAs) could be relatively easy to enforce, serve as broodstock sanctuaries that benefit fished populations, and optimize use of the seafloor (*Fowler et al., 2018*). Moreover, the economic impact of artificial reefs also appears high relative to unstructured control sites. Although the potential positive relationships of artificial reefs with fish populations are encouraging, it is important to recognize the attraction effect of artificial reefs and how such a process can facilitate overfishing (*Bohnsack, 1989*; *Smith, Lowry & Suthers, 2015*, and references therein). Artificial reefs, however, can also enhance growth and survival, for example by providing hard substrate for sessile fouling communities that provide food to fish (*Todd, Lavallin & Macreadie, 2018*). Quantitative evidence may also guide modeling efforts to assess the role of artificial reefs on population dynamics of exploited species. Lastly, given that unstructured seafloor (*i.e.,* flat, soft-bottom habitat) can be many orders of magnitude greater in areal footprint than artificial reefs, it is important to recognize that unstructured habitats can serve as effective juvenile habitat (*sensu Dahlgren et al., 2006*) for exploited fishery species. Unstructured habitat is especially important for species for which ICES gives advice. For example, in comparing coastal habitat use among various structured (*e.g.*, seagrass, kelp, marsh) and unstructured habitats, subtidal soft-bottom was the habitat used as a spawning or nursery area by the largest proportion of ICES species examined, with intertidal soft-bottom habitat as the next most important habitat (*Seitz et al., 2014*).

## Habitat heterogeneity enhances biodiversity

Overall, both natural and artificial hard-bottom or reef structures tend to have a high attraction potential and house densities of fish superior to surrounding habitats (*e.g.*, *Bergström, Sundqvist & Bergström, 2013*; *Støttrup et al., 2014*; *Whitmarsh et al., 2008*). In one study, natural structured habitats (rock, shell, rubble) compared to bare sand had positive effects on density, growth, and survival of juvenile fishes and invertebrates, with a greater positive effect for arthropod invertebrates than vertebrates (*Lefcheck et al., 2019*). When man-made structures are placed on surrounding soft-bottom areas (*e.g.*, wind turbine foundations, oil platforms), they represent beacons of habitat heterogeneity that are capable of attracting new residents (*Gascon & Miller, 1982*; *Stenberg et al., 2015*; *Castège et al., 2016*; *Fujii, 2016*). For example, cod appeared to have a clear affinity for hard-bottom substrates, with increased fish densities and/or biomass near natural hard-bottom reefs (*Stål, Pihl & Wennhage, 2007*; *Støttrup et al., 2017*; *Kristensen et al., 2017*), offshore wind farms (*Bergström, Sundqvist & Bergström, 2013*; *Reubens et al., 2013*; *van Hal, Griffioen & van Keeken, 2017*), oil-platforms (*Fujii, 2015*), shipwrecks (*Wieland et al., 2009*; *Krone et al., 2013*) and other man-made structures (*Langhamer & Wilhelmsson, 2009*; *Wehkamp & Fischer, 2013*). Similarly, pollock generally restricted their movement to a submerged reef area in Loch Ewe, Scotland, while saithe ranged more widely during daytime and returned to the reef during night-time (*Sarno, Glass & Smith, 1994*). Specific age groups of cod and pouting (*Trisopterus luscus*) can have high residency and site fidelity to the

bases of marine wind turbines, feeding on the dominant epifaunal prey growing on these introduced habitats (*Reubens et al., 2014*; *Reubens, Degraer & Vincx, 2014*).

Different species react differently to the addition of hard-bottom substrates, and recent literature reports on various outcomes even within a species. For example, whiting can occur in relatively high abundance on hard substrates (*e.g.*, shipwrecks and offshore wind farms; *Krone et al., 2013*; *Lindeboom et al., 2011*; *Pihl & Wennhage, 2002*), as well as on soft substrates (*Stål, Pihl & Wennhage, 2007*; *van Hal, Griffioen & van Keeken, 2017*). Additionally, in some cases, the attraction of reef-associated demersal fish species such as bogue (*Boops boops*), horse mackerel, and axillary sea-bream may make them more vulnerable to predation, as the probability of prey–predator encounters increases compared to unstructured habitats (*Leitão et al., 2008*). Lastly, some commercially important species, such as dab (*Limanda limanda*) and sole, may avoid rocky substrates, with decreasing abundances whenever there is a shift from soft to hard bottom (*van Hal, Griffioen & van Keeken, 2017*). Thus, the relationship of hard-bottom structures with fish distribution and abundance patterns in the North Sea and Baltic Sea are species-dependent, a factor that must be considered during management and restoration actions.

## Improved fitness and reproduction capacity

Selection by fish for complex, hard-bottom habitats generally improved fitness in those species occupying these habitats. Enhanced foraging opportunities improved condition in cod and pouting when occupying wind farm areas (*De Troch et al., 2013*). However, this is not always the case. For example, *Mathers, Houlihan & Cunningham (1992)* examined growth rates in saithe using biochemical indicators and found no differences in growth rates on an oil platform as compared to the open sea. Further, while pouting condition was enhanced in a wind farm area relative to sandy areas, condition of cod was similar in both habitat types (*Reubens, Degraer & Vincx, 2014*). The similar growth rates or condition of fish on oil platforms or wind farms compared to unstructured areas suggests that although production was evident on a local scale (*i.e.,* wind farm foundations), this may have had little effect on the larger spatial scale (*i.e.,* the Belgian part of the North Sea). In a related example, oil and gas platforms in the North Sea provided unique feeding grounds for commercially important gadoids such as cod, saithe, and haddock (*Fujii, 2016*). For example, haddock predominantly consumed ophiuroids that occurred in high abundances on platforms at all depths, whereas temporal changes in the presence and abundance of saithe reflected the occurrence and availability of euphausiids (*Fujii, 2016*). These findings indicate that gadoids utilized the platform for foraging.

A tagging study revealed that spawning activities of cod in the Gulf of Maine occurred around specific bottom features, such as humps and ridges (*Siceloff & Howell, 2013*). Cod tended to move in groups after spawning, as 14 of 26 tagged and recaptured cod shifted to deep water (>100 m) within 23 to 50 days after tagging, with females tending to leave before males. While these findings indicate that the pelagic spawning of cod may be associated with large offshore hard-bottom structures, relatively shallow, complex habitats with rocks and cobble may enhance survival among the juveniles after benthic settling (*Tupper & Boutilier, 1995*).

A recent study highlighted the potential role of offshore oil and gas platforms as spawning sites for lumpsucker (*Cyclopterus lumpus*; *Todd, Lavallin & Macreadie, 2018*). *C. lumpus* was observed during different stages of reproduction on an offshore oil and gas structure in the North Sea, providing novel evidence of spawning offshore at depths of around 40–45 m. Examples from the literature also show positive relationships of hard-bottom habitats with spawning and recruitment for several fish species, with much of the relevant literature coming from natural hard-bottom habitats. In a survey of natural hard bottoms in western Norway, spawning of herring generally occurred on hard-bottom substrates down to a depth of 10 m, with little evidence of spawning on soft-bottom habitats (*Johannessen, 1986*).

In the Baltic Sea, the vegetation usually attached to hard substrates seems to play an important role for successful spawning of Baltic herring (*C. harengus membras*). In a survey of natural hard bottoms in the northern Baltic Sea (Sweden), herring eggs were only found on filamentous algae (*Pilayella littoralis*) attached to hard substrates (*Aneer & Nellbring, 1982*) and not on the surrounding hard or soft bottoms. Likewise, in the southwestern archipelago of Finland, Baltic herring spawning beds occurred in areas with vegetation growing on hard bottoms (down to a depth of 4 m); usually close to the regions' deepest zones (down to 60 m depth; *Kääriä et al., 1997*). The complex habitat established by the algae may provide refuge from predators, as birds and other fish are responsible for a majority of the predation of herring eggs (*Aneer & Nellbring, 1982*). Areas without spawning beds generally had soft sediments with narrow bands of vegetation. Filamentous green algae (*Cladophora* sp.), brown algae (*Ectocarpus* sp., *Pilayella* sp.) and *Potamogeton pectinatus* made up the most common vegetation where divers observed eggs (*Kääriä et al., 1997*). For Baltic herring spawning on the Lithuanian coast (eastern coast of the Baltic), eggs were found on red algae *Furcellaria lumbricalis* and *Polysiphonia fucoides*, as well as hard habitat covered with mussels at 4–8 m depth (*Šaškov et al., 2014*). The dependence of Baltic herring on vegetation as substrate for the eggs may render this species vulnerable to pressures such as eutrophication that negatively affect macroalgal growth.

Modeling studies can be informative in assessing the effects of habitat on recruitment success. One model was used to assess the effects of MPAs on juvenile cod recruitment success (*Lindholm et al., 2001*). MPAs can maintain habitat complexity (*i.e.,* more pebble-cobble substratum) compared to areas that are fished with dredges or trawls that damage the seafloor and reduce complexity (*Ocean Studies Board & National Research Council, 2002*). Modeled survival of juvenile cod on natural reefs showed that survival was density-dependent (*i.e.,* affected by predation or cannibalism; *Wikan & Eide, 2004*) and increased with seafloor habitat quality (modeled as increased size of an MPA). Juvenile cod had higher survival rates at low densities and in large MPAs, thus linking juvenile cod survival to seafloor habitat complexity (*Lindholm et al., 2001*). Modeled improved survival associated with complex sea-floor habitats is consistent with previous laboratory studies and may explain why various fish species prefer habitats with gravel and cobble where they can find shelter in the crevices (*Gotceitas & Brown, 1993*; *Christoffersen et al., 2018*).

The review process that we used did not include all commercially important species, but rather was targeted towards species important for management by ICES, and we

only examine temperate, non-biogenic reefs. The results from our study are limited to those specific species and the physical structure provided by hard-bottom, non-biogenic habitats. Our search terms were very specific and we acknowledge that this approach may have missed relevant research from habitats that we excluded. Therefore, interpretations of these data apply only to the species and habitats we incorporate.

## Recent case studies

Although the literature for this study was not reviewed beyond those articles published in early April 2017, a related study reviewing methods that have examined the nursery role of different habitats included search terms relevant to hard-bottom habitats and fishery species of interest to ICES (Ciotti et al., in prep.). We highlight several recent case studies below that reflect this updated literature review. In the first example (*Elliott et al., 2017*), stereo baited remote underwater video (SBRUV) surveys were conducted in the nearshore, Scottish waters of the Firth of Clyde during summer 2013 and 2014 to determine the habitat of juvenile Atlantic cod, haddock and whiting. The fishery stocks of all three species declined in the late 20th century, and recruitment and spawning stock biomass remains relatively low in the Firth of Clyde despite efforts to re-build the stocks (*Fernandes & Cook, 2013*; *Holmes et al., 2014*; *International Council for the Exploration of the Sea , 2016a*; *International Council for the Exploration of the Sea , 2016b*; *International Council for the Exploration of the Sea , 2016c*). Understanding juvenile gadoid habitat is particularly important given that settlement and post-settlement survival is thought to be the best means to understanding gadoid population regulation (*Olsen & Moland, 2011*; *Laurel, Knoth & Ryer, 2016*). Atlantic cod were most abundant in shallow, sheltered areas composed of gravel −pebble containing maerl unattached coralline algae. Haddock and whiting predominated over deeper sand and mud. Relative abundances of Atlantic cod and whiting were positively related to the diversity of epibenthic and demersal fauna. Thus, spatial conservation measures to benefit demersal fish should consider habitat type and diversity (*Elliott et al., 2017*).

Another case study examined the environmental benefits of leaving de-commissioned offshore infrastructure, such as oil and gas platforms and wind turbines, in the ocean due to benefits such as biodiversity enhancement, provision of reef habitat, and protection from bottom trawling (*Fowler et al., 2018*). The rationale for this study was that the *in situ* ecosystem value of platforms and the negative impacts of removal are not factored into decommissioning decisions in regions such as the North Sea, where over 80% of oil structures are more than a decade old (*OSPAR Commission, 2017*) and are likely integrated to at least some extent into existing ecosystems (*Fowler et al., 2018*). The removal policy is based on the assumption that "leaving the seabed as you found it" will minimize negative impacts on the marine environment. A total of 200 experts spanning academia, government, and private organizations were sent surveys seeking their opinions on the following: (1) appropriateness of the current removal policy, (2) identification of viable alternatives to complete removal of offshore infrastructure, (3) key environmental considerations and trade-offs for decommissioning decisions, and (4) advice on decommissioning considerations between platforms and wind turbines. Of most relevance to this review

of hard-bottom habitats, 78% of the 40 respondents thought that artificial habitats with environmental value should be maintained and protected due to enhanced ecosystem services (*Fowler et al., 2018*). Evidence includes harboring threatened species (*Bell & Smith, 1999*), providing reef habitat (*Coolen et al., 2020*), boosting recruitment of overfished species (*Love et al., 2006*), producing fish biomass at a greater rate than any other marine ecosystem (by as much as a factor of ten; (*Claisse et al., 2014*)), and acting as foraging sites for top-order predators (*Todd et al., 2009*). *Fowler et al. (2018)* concluded that the traditional view that artificial structures must be removed from marine ecosystems simply because they do not "belong" there has shifted to one of environmental optimization based on comparative assessment (*Fowler, Macreadie & Booth, 2015*). Each decommissioning option will have positive and negative impacts that must be carefully weighed, while also accounting for site-specific characteristics and the broader environmental context of the disturbance.

## CONCLUSIONS AND IMPLICATIONS FOR MANAGEMENT

Our systematic review found that Atlantic cod, the most frequently studied species of the group of species managed by ICES, had increased biomass, density, feeding, and spawning associated with hard-bottom habitats compared to unstructured habitats. Spawning of herring increased on hard-bottom habitats, which appear to be of importance for this species. We detected that most research was conducted on Atlantic cod, herring, and saithe, and the most commonly studied variables included biomass, density, biodiversity, and feeding. Cod was the species most commonly studied, and studies were mostly on community-related variables (*i.e.,* density, biodiversity, biomass). However, not all responses to the presence of hard-bottom habitat were positive. For herring, saithe, striped red mullet, whiting, pollock, plaice, and sole, we quantified positive and negative relationships between fish density and hard-bottom habitats. Thus, we achieved our goal of summarizing the existing information on ICES-managed fish species and their use of temperate hard-bottom habitats. This information should help inform scientists interested in the ecological function of hard-bottom habitats for some key fish species, and inform managers and conservation groups focused on coastal habitats of the potential importance of hard-bottom habitats to certain fish species.

 Natural hard-bottom reefs, if removed or destroyed, can only be restored by human intervention. Thus, it would be wise if management focuses on protecting natural reefs and finding the most cost-effective solutions for restoring reefs. In support of ecosystem-based fisheries management, research should provide additional scientific information on the value of hard-bottom habitats for biodiversity or for individual species on a local, national and regional scale. While this information is slowly being produced (*e.g., Christoffersen et al., 2018*; *Liversage & Benkendorff, 2013*; *Liversage et al., 2017*), the economic value of hard-bottom habitats remains largely unknown. This is in contrast to other complex marine habitats (*e.g.,* eelgrass) where several recent studies have estimated the annual economic values per area (*McArthur & Boland, 2006*; *Tuya, Haroun & Espino, 2014*; *Cole & Moksnes, 2016*). Similar economic valuation studies are needed for hard-bottom habitats,

so that policy makers may assess the economic importance of protective measures related to these habitats. In areas where hard-bottom habitat has been removed or destroyed, consideration should be given to the economic and ecological benefits or disadvantages of removing decommissioned marine structures (*e.g.*, *Fowler et al., 2018*). These structures could be equated to shipwrecks that become *de facto* reefs and are generally not removed.

The value of reef habitats and the need for their protection ultimately depends on the occurrence of that habitat type in the area/region and the potential for hard-bottom habitats to represent value as a resource. For example, in Denmark, although the extraction of boulders and larger stones from reef areas has ceased, extraction of smaller stones or gravel continues to take place, even though these hard bottom types are important habitats for juvenile eel and cod (*Gotceitas, Fraser & Brown, 1995*; *Christoffersen et al., 2018*).

Research should provide the necessary information to allow for an effective restoration of the functions of destroyed reefs. Studies are increasingly investigating restoration methodologies for hard-bottom reefs (*Støttrup et al., 2017*; *Liversage & Chapman, 2018*). However, current information is insufficient to conduct a restoration program that is not, in itself, explorative. From a cautionary standpoint, we recommend that anthropogenic activities degrading hard-bottom habitats should be carefully considered and weighed against the accumulating evidence that highlights the importance of various hard-bottom habitats for several commercially important species.

## ACKNOWLEDGEMENTS

This work stemmed from an ICES working group on the Value of Coastal Habitats for Exploited Species, and we thank all participants of the working group for discussions leading to insights incorporated in this research.

### Funding

Hugo Flávio was supported by COST (European Cooperation in Science and Technology) Action 15121 "Advancing marine conservation in the European and contiguous seas" (*Katsanevakis et al., 2017*), by the Horizon 2020 Framework Programme for Research and Innovation, and by the Ministry of Environment and Food of Denmark through the Danish Marine Coastal Fisheries Management Program (Marin Fiskepleje). Funding to David Eggleston was provided by NSF grant OCE-1155609 and NC State University. Jon C. Svendsen was supported via projects funded by (1) European Maritime and Fisheries Fund and the Danish Fisheries Agency (33113-B-16-057 and 33113-B-19-142), (2) the Danish Rod and Net Fish License Funds (39 133), (3) the Velux Foundation, (4) Vattenfall and (5) the EU Interreg project MarGen. Josianne Støttrup was supported by the project "Importance of reef habitats for fish, harbour porpoise and fisheries management" (33113-B-16-057) funded by the European Maritime and Fisheries Fund. Support for all co-authors' travel to ICES working group meetings was provided by the National Institute of Aquatic Resources, Technical University of Denmark, MarCons, the Virginia Institute

of Marine Science, and North Carolina State University. There was no additional external funding received for this study. The funders had no role in study design, data collection and analysis, decision to publish, or preparation of the manuscript.

## Grant Disclosures

The following grant information was disclosed by the authors:
European Cooperation in Science and Technology): Action 15121.
Danish Marine Coastal Fisheries Management Program.
NC State University: OCE-1155609.
Danish Fisheries Agency: 33113-B-16-057, 33113-B-19-142.
Danish Rod and Net Fish License Funds: 39 133.
Velux Foundation.
Vattenfall.
EU Interreg project MarGen.
European Maritime and Fisheries Fund: 33113-B-16-057.
National Institute of Aquatic Resources.
Technical University of Denmark, MarCons.
Virginia Institute of Marine Science.
North Carolina State University.

## Competing Interests

The authors declare there are no competing interests.

## Author Contributions

- Hugo Flávio conceived and designed the experiments, performed the experiments, analyzed the data, prepared figures and/or tables, authored or reviewed drafts of the article, and approved the final draft.
- Rochelle Seitz conceived and designed the experiments, performed the experiments, authored or reviewed drafts of the article, and approved the final draft.
- David Eggleston conceived and designed the experiments, performed the experiments, authored or reviewed drafts of the article, and approved the final draft.
- Jon C. Svendsen performed the experiments, authored or reviewed drafts of the article, and approved the final draft.
- Josianne Støttrup conceived and designed the experiments, performed the experiments, analyzed the data, prepared figures and/or tables, authored or reviewed drafts of the article, and approved the final draft.

## Data Availability

The code in the R script was used to develop results and comparisons among various hard-bottom habitats and economically important species and is available in the Supplementary File.

## Supplemental Information

Supplemental information for this article can be found online at http://dx.doi.org/10.7717/peerj.14681#supplemental-information.

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
