# Peer review of "Hard-bottom habitats support commercially important fish species: a systematic review for the North Atlantic Ocean and Baltic Sea"

_PeerJ, doi:10.7717/peerj.14681_

## Round 0.1 · original submission · Minor Revisions

Two reviewers provided a thoughtful and comprehensive review of this manuscript, which I agree is a nice review of the literature. Some minor suggestions for wording and style are provided by the reviewers, though I encourage the authors to address the concern that the literature search was performed 5 years ago in 2017 - surely there would be more data available in the literature to include in the systematic review. If not included, what gaps may exist in this review since 2017?

A few other concerns regard the keywords used in the search - excluding terms such as 'demersal' and 'northwest Atlantic', as well as the relatively limited taxonomic scope (but focused on important fisheries under ICES) should be addressed in a rebuttal and perhaps discussion in the paper.

If the authors can address these concerns I think this will be a valuable contribution to our understanding and management of various fish habitats.

Reviewer 1 ·

Basic reporting

Overall, the paper is well written and readable. There are a few minor errors, as listed below:
- Line 94 manged = managed
- Line 118 feces = faeces /pseudofaeces
- Line 124 sreening = screening
- Line 469 ICES-manged = ICES-managed (might be a couple more of these, so conduct a search)
There a couple of sentences that start with “This…” e.g. line 199. I always find “This..” vague and it would be worthwhile revising the start of those sentences to be more specific.
Line 120 refers to “conducting the search”, would be clearer to instead state “search protocol”.
Lines from 301 refer to reefs generally, but as you exclude biogenic reefs I feel that you have to be more specific in this section when discussing reefs.

Experimental design

The literature review is comprehensive and the criteria for inclusion/exclusion are laid out clearly in the methods, as is the research question.
The main issue with the review is that the search was conducted in April 2017. My feelings are that the gap between 2017-2022 should be filled prior to publication, as there may have been some pertinent papers published since.
Could demersal be included as a search term?
Could there be some additional ecosystem-related keywords included? E.g. boulders, bedrock, MDAC, outcroppings etc to see if those also provide some additional studies?
Line 135 refers to some studies being missed. How can exclusion of papers be mitigated? Could the authors conduct additional or manual searches?
There is no mention anywhere in the review of depth. It would be nice to know max depths of the studies included in the review, as depth is vital to species distribution.

Validity of the findings

The findings are valid and confirm already known trends and themes. It is nice to have the findings gathered together in one place in a well written, systematic review. I feel that if the authors can fill in the blanks since 2017 and also add in some additional search terms, they could find some additional papers to provide a fully rounded review. However, I suspect that the additional papers, won't affect the overall take-home messages and themes of the review paper.

Additional comments

Overall, an excellent, easy to read, paper, and one I would suggest is suitable for publication, once the above minor issues are addressed. The figures are straightforward and easy to understand.

Reviewer 2 ·

Basic reporting

The manuscript is written in clear, unambiguous English. The background presented in the Introduction is sufficient to provide a context for the literature review that follows. The supporting citations are all appropriately referenced. The figures and tables were all labelled and their titles were informative.
Introduction – good rationale: importance of natural hard bottom substrate (temperate reefs) for fish populations – what about artificial substrates and how these might impact management decisions.

Experimental design

The design is a literature review of literature examining the interaction between commercially important species and natural or artificially developed hard substrates. Sources are adequately cited (paraphrased) and there is logical flow

Validity of the findings

Custom checks: relevant and meaningful, robust and believable, Yes
The authors are clear in explaining how they winnowed an apparently large (>2400 papers) body of literature down to 45 papers that were relevant to their study. Because they restricted their interest to ICES managed fish species, most of their findings were focused on a very few species (cod, herring, pollack/saithe). I would have liked to see more species included, especially those populations that are of concern and have life histories that might make artificial substrates beneficial, for example, lumpfish, wolffishes, pouts, rockfishes. They did touch on one study that examined lumpfish distributions on a fixed oilrig. It would also have been interesting to see invertebrates such as lobster included. Although Canada and United States were keywords, they did not include the northwest Atlantic, so one cannot determine if there is little relevant literature for the western side of the Atlantic, or if it was not considered. This is a minor criticism of the limited but credible objectives of an ICES working group that they would restrict their scope mainly to the northeast Atlantic and its commercially relevant species.

Additional comments

Minor corrections (by line number):
94 – ICES-managed
Also, the acronym ICES was explained in the abstract but not in the body of the paper
108 – for (not fore)
124 – screening
126 – perhaps there is a verb missing as in the ‘first author was screened by title’?
143 – conducted
Missing some can Atlantic species of interest – white hake, wolffish, lumpfish, and non fish (lobster)
228 – were excluded for “other” reasons
238 – state latin names for herring and saithe here
242 – analyzed or analysed, move Pollachius virens to line 238
247 – move Clupea harengus to line 238
300 – is this Reubens et al 2014a?
319 - habitats
444 – use ‘survival was density-dependent’ to be consistent with the verbal tense of the sentence
469 – ICES-managed
512 – inconsistent style with other references
514 – inconsistent style
516 - inconsistent style
532 - inconsistent style
542 – is there additional information on the publishing agent
564 – move Hartman to its own line. Should there be a period between California and Bight?
575 – ctrl-F did not show a Katsanevakis in the text
588 – is format of reference correct?
595 – inconsistent style (capitalized words in title)
598 – inconsistent style (capitalized words in title)
615 – Pollachius virens in italics
635 - inconsistent style (capitalized words in title)
682 - inconsistent style (capitalized words in title)

---

## Round 0.2 · accepted · Accept

The authors have done an excellent job of addressing some minor reviewer concerns and this paper can now be accepted for publication. Congratulations!

Reviewer 2 ·

Basic reporting

accepted

Experimental design

accepted

Validity of the findings

accepted

Additional comments

The authors have prepared a detailed response to my review of their draft. I accept their revisions, including their arguments for leaving the text as is, and recommend that acceptance of the manuscript. thank you for the opportunity to read the revised manuscript and rebuttal.